# Intermittent hypoxia mediated by TSP1 dependent on STAT3 induces cardiac fibroblast activation and cardiac fibrosis

Qiankun Bao[1]*, Bangying Zhang[1], Ya Suo[1], Chen Liu[2], Qian Yang[1], Kai Zhang[1], Ming Yuan[1], Meng Yuan[1], Yue Zhang[1], Guangping Li[1]*

[1]Tianjin key laboratory of Ionic-Molecular Function of Cardiovascular Disease, Department of Cardiology, Tianjin Institute of Cardiology, The Second Hospital of Tianjin Medical University, Tianjin, China; [2]Department of Clinical Laboratory, Peking University People's Hospital, Beijing, China

**Abstract** Intermittent hypoxia (IH) is the predominant pathophysiological disturbance in obstructive sleep apnea (OSA), known to be independently associated with cardiovascular diseases. However, the effect of IH on cardiac fibrosis and molecular events involved in this process are unclear. Here, we tested IH in angiotensin II (Ang II)-induced cardiac fibrosis and signaling linked to fibroblast activation. IH triggered cardiac fibrosis and aggravated Ang II-induced cardiac dysfunction in mice. Plasma thrombospondin-1 (TSP1) content was upregulated in both IH-exposed mice and OSA patients. Moreover, both in vivo and in vitro results showed IH-induced cardiac fibroblast activation and increased TSP1 expression in cardiac fibroblasts. Mechanistically, phosphorylation of STAT3 at Tyr705 mediated the IH-induced TSP1 expression and fibroblast activation. Finally, STAT3 inhibitor S3I-201 or AAV9 carrying a periostin promoter driving the expression of shRNA targeting Stat3 significantly attenuated the synergistic effects of IH and Ang II on cardiac fibrosis in mice. This work suggests a potential therapeutic strategy for OSA-related fibrotic heart disease.

*For correspondence:
baoqiankun@tmu.edu.cn (QB);
tic_tjcardiol@126.com (GL)

**Competing interests:** The authors declare that no competing interests exist.

## Introduction

The prevalence of sleep apnea is high in patients with cardiovascular disease, which could be caused by multiple risk factors including respiratory instability, obesity and upper airway dysfunction (*Javaheri et al., 2017*; *Kasai et al., 2012*). Sleep apnea, classified as obstructive sleep apnea (OSA) and central sleep apnea, affects or covaries with numerous health outcomes and physiological processes, particularly cardiovascular disease (*Javaheri et al., 2017*; *Kasai et al., 2012*; *Somers et al., 2008*). A recent prospective study of OSA found it associated with increased risk of incident heart failure in community-dwelling middle-aged and older men (*Gottlieb et al., 2010*). OSA is associated with a profile of perturbations that include sympathetic nerve activity, metabolic dysregulation, inflammation, oxidative stress, vascular endothelial dysfunction, and intermittent hypoxia (IH), all of which are critical for the pathogenesis of coronary heart disease, hypertension, atrial fibrillation, and heart failure (*Javaheri et al., 2017*; *Kasai et al., 2012*). IH is a hallmark of OSA and initiates several mechanisms that lead to cardiac fibrosis and cardiac dysfunction (*Baguet et al., 2012*; *Wei et al., 2016*).

Cardiac fibrosis is characterized by excessive deposition of extracellular matrix proteins by cardiac fibroblasts (CFs), which respond to pathological stress and environmental stimuli by transforming into myofibroblasts. Activated CFs express elevated levels of various proinflammatory and profibrotic factors causing fibroblast proliferation, migration and scar formation (*Travers et al., 2016*). The persistence of myofibroblasts eventually results in adverse changes in ventricular structure and

compliance, thereby leading to heart failure (*Baguet et al., 2012*; *Travers et al., 2016*). However, the effect and mechanism of CF activation in IH-induced cardiac remodeling remain unclear.

One central cytokine linked to cardiac fibrosis is transforming growth factor β (TGFβ), which is highly regulated at the level of activation (*Travers et al., 2016*; *Meng et al., 2016*). The activation of latent pro-TGFβ requires thrombospondin-1 (TSP1, encoded by *Thbs1* gene), which is a matricellular glycoprotein and can be secreted by various cell types, to remove its latency-associated pro-peptide (*Meng et al., 2016*; *Crawford et al., 1998*; *Adams and Lawler, 2011*). Myocardial TSP1 expression was increased in a mouse model of pressure overload because of transverse aortic constriction (*Xia et al., 2011*), and blocking TSP1-dependent TGFβ activation prevented cardiac fibrosis progression and improved cardiac function (*Belmadani et al., 2007*). However, the role and underlying mechanism of TSP1 in IH-induced CF activation and cardiac fibrosis remain to be elucidated.

As a member of the signal transducer and activator of transcription (STAT) protein family, STAT3 was originally identified as an interleukin-6–activated transcription factor. It can also be phosphorylated by receptor-associated Janus kinase (JAK) in response to growth factor and hemodynamic stress, thus acting as a regulator in fundamental cellular processes including inflammation, cell growth, proliferation, differentiation, migration, and apoptosis (*Wei et al., 2003*; *Chakraborty et al., 2017*; *He et al., 2018*). Emerging evidence demonstrates that STAT3 signaling is hyperactivated in fibrotic diseases, which may be an important molecular checkpoint for tissue fibrosis (*Chakraborty et al., 2017*; *Su et al., 2017*). Recent study demonstrated that STAT3 can drive TSP1 expression in astrocytes (*Tyzack et al., 2014*). Given the integrated function of STAT3 activation in inflammation and fibrosis, we hypothesized that IH-induced STAT3 activation might play a crucial role in CF activation and cardiac fibrosis by increasing TSP1 expression.

In the present study, we investigated the effect of IH exposure on cardiac fibrosis in response to angiotensin II (Ang II) in mice and the potential underlying mechanism. TSP1 expression induced by IH in CFs, mediated by phosphorylation of STAT3 at Tyr705, was involved in CF activation and cardiac fibrosis. Pharmacological or genetic inhibition of STAT3 restrained IH-induced CF activation and cardiac fibrosis and ameliorated IH-induced cardiac dysfunction.

## Results

### IH induced cardiac fibrosis and aggravated Ang II-induced cardiac dysfunction in mice

Most respiratory events of patients with OSA result in desaturation–reoxygenation sequences that cause IH (*Baguet et al., 2012*). To investigate IH exposure to cardiac function, we housed mice under IH or normoxia for 28 days (*Figure 1A*). Hypoxia in heart tissue was evaluated by using pimonidazole (*Figure 1—figure supplement 1A*). IH exposure slightly increased the ratio of heart weight to tibial length (*Figure 1B*). Echocardiography analysis revealed a moderate decrease in ejection fraction (EF) and fractional shortening (FS) with IH as compared with normoxia (*Figure 1C–D*). Furthermore, Masson and Sirius red staining demonstrated mildly larger fibrosis area in the heart of mice after IH exposure (*Figure 1E–F*).

To investigate the effect of IH associated with cardiac function after injury, we used the mouse model of Ang II-induced cardiac hypertrophy to represent cardiac fibrotic responses. Consistent with previous studies (*Su et al., 2017*; *Schafer et al., 2017*), Ang II treatment induced cardiac fibrosis with increasing the ratio of heart weight to tibial length and fibrosis area and impaired EF and FS (*Figure 1B–D*). Strikingly, IH exposure further impaired cardiac function, with increasing the ratio of heart weight to tibial length and decreasing EF and FS (*Figure 1B–D*). IH exposure significantly enlarged the fibrotic area of heart as well (*Figure 1E–F*). IH exposure had little effect on myocyte size or ratio of apoptotic cells in the left ventricle with or without Ang II challenge (*Figure 1—figure supplement 1B–C*). These results indicated that IH could induce cardiac fibrosis both at the basal level and in response to Ang II and also aggravated Ang II-induced cardiac dysfunction.

### IH induced fibroblast activation in myocardial interstitium

CFs are now recognized for their fundamental contributions to the heart's response to various forms of injury (*Travers et al., 2016*). To investigate the effect of IH on fibroblast activation, we analyzed the expression of α-SMA, collagen I and periostin in the left ventricle of mice after IH exposure. First,

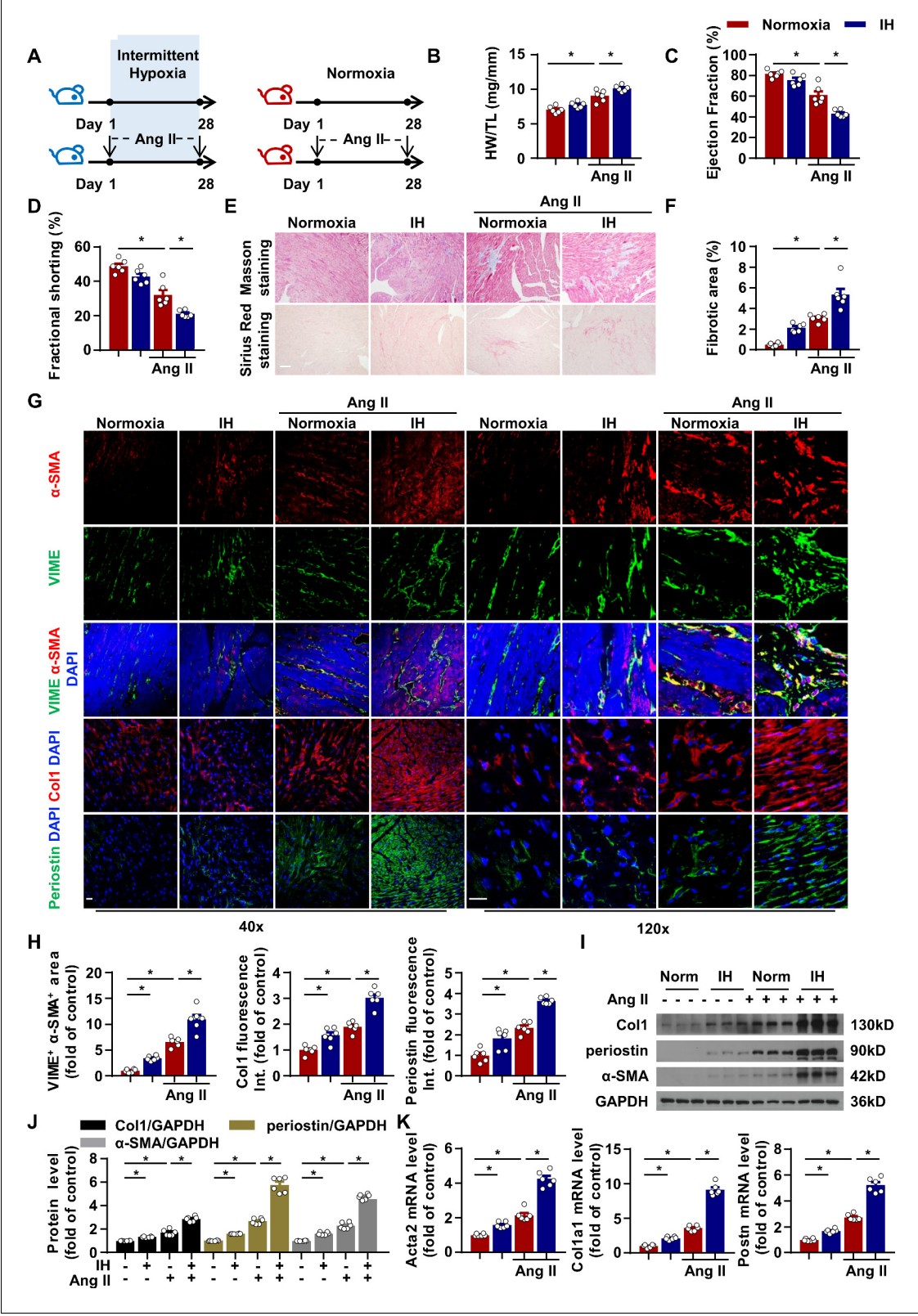

**Figure 1.** Intermittent hypoxia (IH) induces cardiac fibrosis and aggravates pathological cardiac dysfunction by promoting fibroblast activation in myocardial interstitium. (**A**) C57BL/6 mice were housed under normoxia or IH with or without infusion of angiotensin II (Ang II) for 28 days. (**B**) Ratio of heart weight to tibial length of mice in each group. (**C, D**) Ejection fraction (EF) and fractional shortening (FS) of mice quantified by echocardiography. (**E**) Representative images of Masson and Sirius red staining of left ventricle. Scale bar, 100 μm. (**F**) Quantification of fibrotic area in (**E**). (**G**)

*Figure 1 continued on next page*

*Figure 1 continued*

Representative confocal microscopy images of immunofluorescence staining for α-SMA, vimentin (VIME, fibroblast marker), collagen I (Col1), periostin and DAPI. Scale bar, 20 μm. (**H**) Quantification of co-localization of α-SMA and VIME, Col1 and periostin fluorescence intensity in (**G**). (**I**) Col1, periostin, and α-SMA protein level in left ventricle detected by western blot analysis. (**J**) Quantification of Col1, periostin and α-SMA protein level in (**I**). (**K**) Quantification of *Acta2, col1a1 and Postn* mRNA expression. Data are mean ± SEM, n = 6 mice per group, *p<0.05, 2-way ANOVA with Bonferroni post-test. Data used for quantitative analyses as well as the numerical data that are represented in graphs are available in *Figure 1—source data 1*. The online version of this article includes the following source data and figure supplement(s) for figure 1:

**Source data 1.** Intermittent hypoxia (IH) induces fibroblast activation and cardiac fibrosis.
**Figure supplement 1.** The effects of IH on cardiomyocyte morphology and apoptotic status.
**Figure supplement 1—source data 1.** The effects of IH on cardiomyocyte morphology and apoptotic status.

immunofluorescence staining showed strongly α-SMA signaling in vimentin-positive CFs in the IH group. IH-induced α-SMA expression in CFs was further enhanced under the Ang II-induced cardiac pathologic condition (*Figure 1G–H*). Meanwhile, the expression of collagen I and periostin showed a similar trend as α-SMA expression (*Figure 1G–H*). In addition, IH exposure slightly increased the mRNA and protein levels of α-SMA, collagen I and periostin (*Figure 1I–K*) in the mouse left ventricle and significantly elevated the levels in the left ventricle of Ang II-infused mice (*Figure 1I–K*). These results suggest that IH can activate CFs at the basal level and under the Ang II-induced injury condition.

## IH exposure promoted TSP1 expression in CFs

To test whether TSP1 plays an important role in the progression of cardiac fibrosis induced by IH, we next detected TSP1 expression in mice after IH exposure. As compared with normoxia, IH exposure significantly increased plasma TSP1 content and mRNA level in the mouse left ventricle, especially in Ang II-infused mice (*Figure 2A–B*). The protein level of TSP1 in the left ventricle showed a similar trend to that of collagen I and periostin (*Figure 2C–D*). Given that TGFβ and its downstream Smad signaling critically modulate the fibroblast phenotype (*Travers et al., 2016*; *Cucoranu et al., 2005*), we assessed Smad2/3 phosphorylation in the mouse left ventricle. The phosphorylation of Smad2/3 was slightly increased after IH exposure but greatly increased after IH exposure in the Ang II-induced injury condition (*Figure 2C–D*), which is consistent with TSP1 expression. Furthermore, immunofluorescence staining showed elevated signaling of TSP1 in the left ventricle after IH, especially in the Ang II-induced condition (*Figure 2E–F*). In addition, the expression of TSP1 in cells positive for Thy1 (an inclusive surface protein given its association with CFs [*Ali et al., 2014*; *Hudon-David et al., 2007*]) and negative for CD11b (macrophage marker), CD31 (endothelial cell marker), CD45 and Ter119 (hematopoietic cell marker) in the heart, detected by flow cytometry, was significantly higher after IH exposure (*Figure 2G*), which indicated that IH exposure could induce TSP1 expression in CFs.

To identify the potential relevance of TSP1 to OSA, we measured the concentration of TSP1 in plasma samples from 21 patients with OSA and 21 healthy subjects. TSP1 content was significantly increased in patients with OSA as compared with healthy individuals (*Figure 2H*). Together, these data suggest that TSP1 level was elevated in both mice and humans after IH exposure and might contribute to CF activation and cardiac fibrosis.

## IH induced primary CF (PCF) activation via TSP1

Next, to confirm the effect of IH on PCF activation in vitro, we used immunofluorescence staining of α-SMA in mouse PCFs (mPCFs) with or without IH exposure. As compared with normoxia, IH increased α-SMA expression in PCFs and led to an activated phenotype of fibroblast cells (*Figure 3A–B*). In addition, IH exposure increased the proliferation and contractility of PCFs (*Figure 3C–D*). We tested whether TSP1 plays an important role in fibroblast activation and found the mRNA expression of *Thbs1* significantly increased by IH as compared with normoxia in mPCFs (*Figure 3E*). In addition, IH significantly increased the protein level of TSP1 beginning from 2 hr and sustained until 8 hr in mPCFs (*Figure 3F–G*). Furthermore, shRNA lentivirus used to produce TSP1 deficiency in PCFs attenuated IH-induced mRNA and protein levels of α-SMA, collagen I and periostin (*Figure 3H–K*). TSP1 deficiency in PCFs also blocked inflammatory gene Tnfa expression

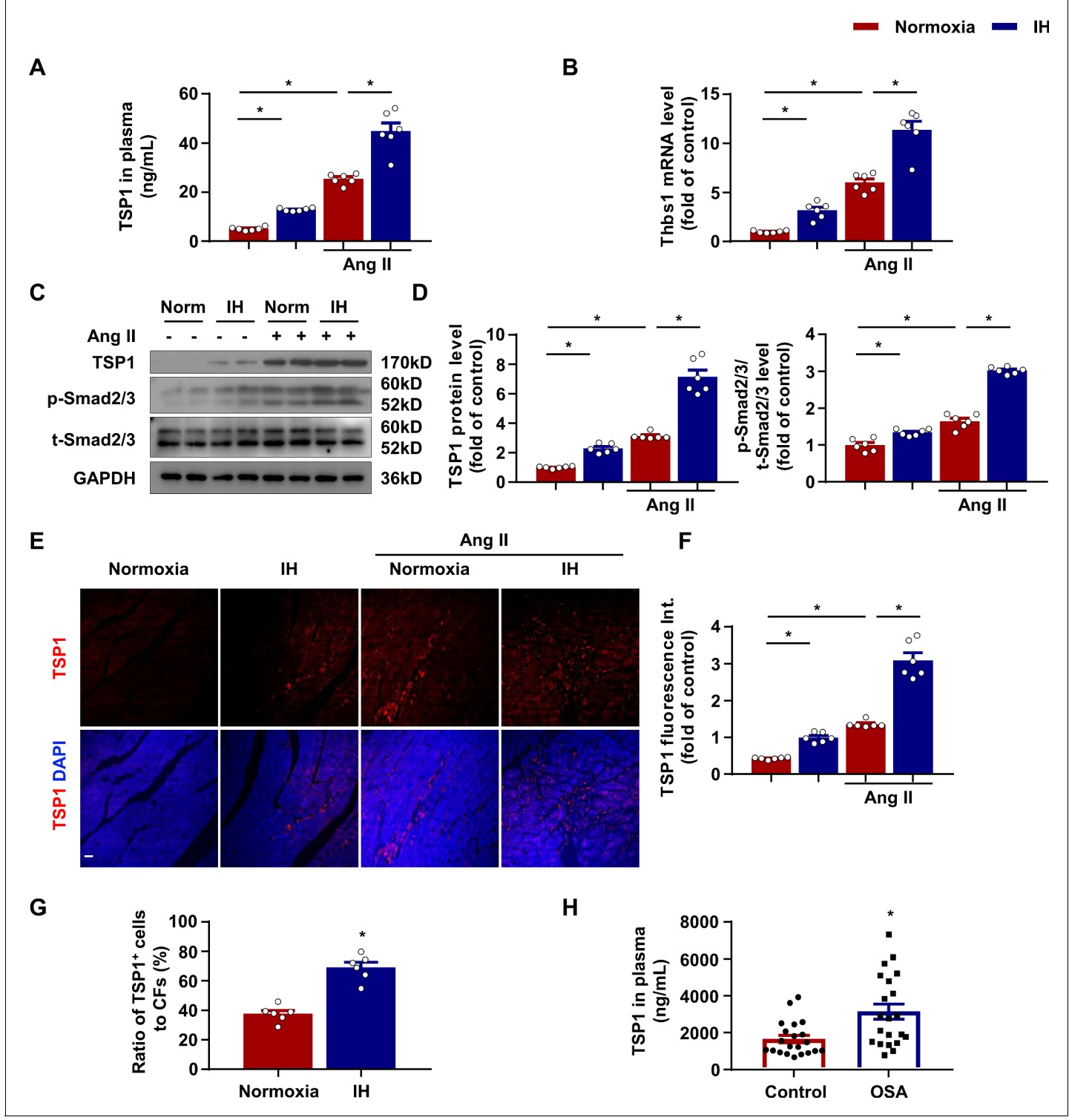

**Figure 2.** Increased thomsbospondin-1 (TSP1) expression in mice after IH exposure. (A) C57BL/6 mice were housed under normoxia or IH with or without infusion of Ang II for 28 days. Plasma concentration of TSP1 in mice detected by ELISA. (B) *Thbs1* mRNA level in left ventricle of mice exposed to IH with or without infusion of Ang II for 14 days quantified by RT-PCR. (C) C57BL/6 mice were housed under normoxia or IH with or without infusion of Ang II for 28 days. The protein levels of TSP1, phosphorylated Smad2/3 (p-Smad2/3), and total Smad2/3 (t-Smad2/3) in left ventricle of mice detected by western blot analysis. (D) Quantification of TSP1 and p-Smad2/3 in (C). (E) Representative confocal microscopy images of immunofluorescence staining for TSP1 and DAPI. Scale bar, 20 μm. (F) Quantification of TSP1 fluorescent intensity in (E). Data are mean ± SEM, n = 6 mice per group, *p<0.05, 2-way ANOVA with Bonferroni post-test. (G) Quantification of number of cells negative for CD45, CD31, CD11b, and Ter119 and positive for

*Figure 2 continued on next page*

*Figure 2 continued*

Thy1 from left ventricle tissue digestion, that stained positive for intracellular TSP1 by flow cytometry. Data are mean ± SEM, n = 6, *p<0.05, unpaired 2-tail *t* test. (H) Plasma concentration of TSP1 in healthy individuals and patients with obstructive sleep apnea (OSA) detected by ELISA. Data are mean ± SEM, n = 21, *p<0.05, unpaired 2-tail *t* test. Data used for quantitative analyses as well as the numerical data that are represented in graphs are available in *Figure 2—source data 1*.

The online version of this article includes the following source data for figure 2:

**Source data 1.** Intermittent hypoxia (IH) increased thomsbospondin-1 (TSP1) expression.

---

(*Figure 3I*) induced by IH exposure. Thus, our results suggested that IH induced PCF activation through upregulating TSP1.

## IH upregulated phosphorylation of STAT3 at Tyr705 site both in vivo and in vitro

TSP1 was reported to be transcriptionally regulated by STAT3 in astrocytes (*Tyzack et al., 2014*), and emerging evidences has indicated that STAT3 involved in fibrosis diseases (*Chakraborty et al., 2017*; *Wang et al., 2018*). Here, we hypothesized that STAT3 might play a vital role in IH-induced TSP1 expression and cardiac fibrosis. Since the phosphorylation status of STAT3 is related to its transcriptional activity (*Darnell et al., 1994*; *Ihle, 1995*), we determined the phosphorylation of STAT3 at multiple sites as well as the phosphorylation of JAKs in mPCFs under IH. STAT3 phosphorylation at Tyr705 was significantly increased by IH from 0.5 to 2 hr, but STAT3 phosphorylation at Ser727 or total STAT3 was not altered by IH exposure (*Figure 4A–B*). Meanwhile, IH exposure significantly induced the phosphorylation of JAK2 at Tyr1008 without affecting that of JAK1 or JAK3 (*Figure 4A–B*). A JAK2 inhibitor, TG101209, abolished the phosphorylation of STAT3 at Tyr705 induced by IH (*Figure 4C–D*), so IH-induced STAT3 phosphorylation might be mediated by JAK2. Phosphorylation of STAT3 at Tyr705 was reported to be crucial to STAT3 nuclear translocation and transcriptional activity (*Darnell et al., 1994*; *Ihle, 1995*), so we next analyzed the STAT3 subcellular localization. STAT3 nuclear translocation was significantly increased by IH on immunofluorescent staining (*Figure 4E–F*). Moreover, in the mouse left ventricle after IH exposure, consistent with results in vitro, IH exposure induced phosphorylation of STAT3 at Tyr705 but not Ser727 (*Figure 4G–H*). Also, IH exposure or together with Ang II markedly increased STAT3 nuclear localization in vivo (*Figure 4I*). Thus, IH induced STAT3 activation via JAK2-mediated STAT3 phosphorylation at Tyr705.

## STAT3 silencing or inhibition blunted IH-induced TSP1 expression and CF activation

We next explored the role of STAT3 in IH-induced TSP1 expression and CF activation. Knockdown of STAT3 with siRNA almost completely abolished IH-induced TSP1 expression both in mRNA and protein level (*Figure 5A–C*). Simultaneously, IH-induced α-SMA, collagen I and periostin expression was blocked by STAT3 knockdown at both the mRNA and protein level (*Figure 5A–C*). To further assess the importance of STAT3 transcriptional activity, we treated PCFs with S3I-201, a selective STAT3 inhibitor. S3I-201 barely showed effect on the basal level of TSP1, α-SMA, collagen I and periostin expression but significantly attenuated IH-induced expression of TSP1, α-SMA, collagen I and periostin at both the mRNA and protein level (*Figure 5D–F*). Quantitative chromatin immunoprecipitation (ChIP) assay proved that IH enhanced the enrichment of STAT3 at TSP1 promoter in PCFs as compared with the control (*Figure 5—figure supplement 1A*). Taken together, our results indicated a predominant role of STAT3 in IH-induced CF activation.

## Pharmacological or genetic inhibition of STAT3 ameliorates IH-induced cardiac dysfunction and fibrosis

As we found that STAT3-induced TSP1 contributed to IH-induced CF activation, we injected S3I-201 into IH-exposed mice with or without Ang II infusion for 28 days (*Figure 6A*). First, echocardiography analysis showed that S3I-201 promoted significant recovery of ratio of heart weight to tibial length (*Figure 6B*) and EF after IH exposure (*Figure 6C*). Moreover, on Masson staining, S3I-201 reduced IH-induced fibrosis under both basal and Ang II infusion conditions (*Figure 6D–E*). In addition, immunofluorescence staining showed significantly decreased TSP1, collagen I, periostin and α-SMA

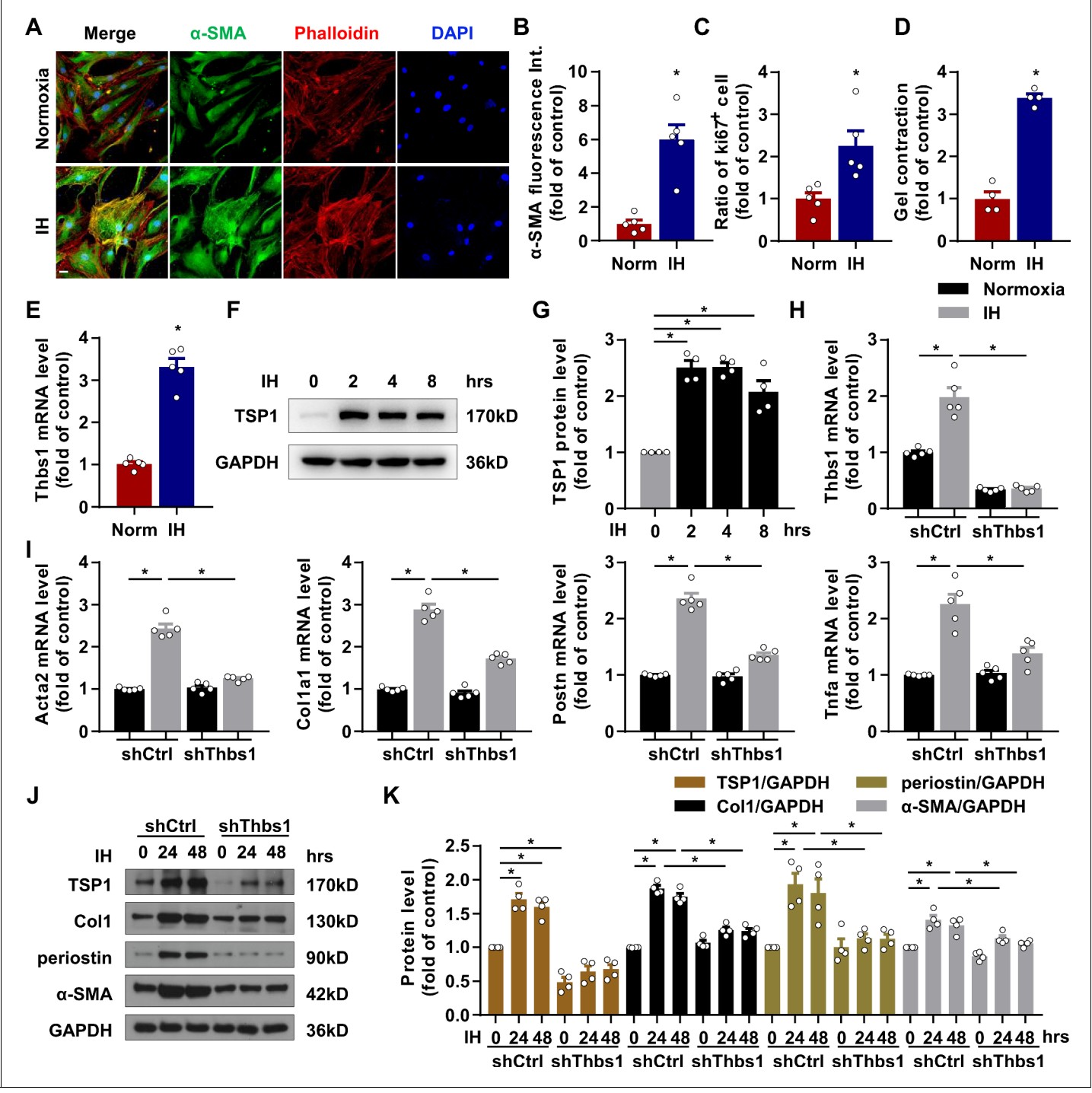

**Figure 3.** IH induced cardiac fibroblast (CF) activation via TSP1. (A) Mouse primary CFs (mPCFs) were cultured under normoxia or IH for 24 hr. Representative confocal microscopy images of immunofluorescence stained for α-SMA, phalloidin and DAPI. Scale bar, 20 μm. (B) α-SMA fluorescence intensity quantification. (C) Ratio of Ki67 positive cells to total mPCFs after IH exposure for 24 hr. Data are mean ± SEM, n = 5 independent experiments, *p<0.05, unpaired 2-tail *t* test. (D) Quantification of gel contraction by mPCFs after IH exposure for 24 hr. Data are mean ± SEM, n = 4 independent experiments, *p<0.05, unpaired 2-tail *t* test. (E) *Thbs1* mRNA levels detected by RT-PCR. Data are mean ± SEM, n = 5 independent experiments, *p<0.05, unpaired 2-tail *t* test. (F) Expression of TSP1 in mPCFs exposed to IH for the indicated time detected by western blot analysis. (G) Quantification of TSP1 in (F). Data are mean ± SEM, n = 4 independent experiments, *p<0.05, 1-way ANOVA with Bonferroni post-test. (H–K) mPCFs were infected with lentivirus to deliver Thbs1 shRNA (LV-shThbs1) or control shRNA (LV-shCtrl) for 48 hr, then cultured under normoxia or IH for 24 hr. (H–I) *Thbs1, Acta2, Col1a1, Postn* and *Tnfa* mRNA levels detected by RT-PCR. Data are mean ± SEM, n = 5 independent experiments, *p<0.05, 2-way ANOVA with Bonferroni post-test. (J) Protein level of TSP1, collagen I (Col1), periostin and α-SMA detected by western blot analysis. (K) Quantification

*Figure 3 continued on next page*

*Figure 3 continued*

of TSP1, Col1, periostin and α-SMA in J). Data are mean ± SEM, n = 4 independent experiments, *p<0.05, 2-way ANOVA with Bonferroni post-test. Data used for quantitative analyses as well as the numerical data that are represented in graphs are available in *Figure 3—source data 1*. The online version of this article includes the following source data for figure 3:

**Source data 1.** Intermittent hypoxia (IH) induced cardiac fibroblast (CF) activation.

expression in the mouse left ventricle after S3I-201 treatment under both basal and Ang II infusion conditions (*Figure 6D–E*). The expression of TSP1, collagen I, periostin and α-SMA, as well as phosphorylation of STAT3 at Tyr705 site in the left ventricle showed similar trends (*Figure 6F–G*).

To further investigate whether S3I-201 could reverse the damage already induced by IH, we treated mice with S3I-201 for 2 weeks after exposure to IH and Ang II for 28 days (*Figure 6—figure supplement 1A*). The ratio of heart weight to tibial length, EF and fibrotic area in mice were comparable between groups (*Figure 6—figure supplement 1B–E*). In addition, because systemic S3I-201 administration might protect the heart via direct and indirect mechanisms, we determined the effect of AAV9-periostin promoter-shStat3 on IH-induced fibrosis (*Figure 6—figure supplement 1F*). Evidence from flow cytometry confirmed the knockdown of STAT3 in CFs (*Figure 6—figure supplement 1G*). Similar to that observed in animals subjected to S3I-201 treatment, AAV9-periostin promoter-shStat3 promoted a significant recovery of ratio of heart weight to tibial length and EF after IH exposure (*Figure 6—figure supplement 1H–I*), and reduced IH-induced fibrosis under both basal and Ang II infusion conditions (*Figure 6—figure supplement 1J–K*). These results suggest that pharmacological or genetic inhibition of STAT3 might be a potential therapeutic strategy for cardiac fibrosis induced by IH.

## Discussion

In the last decades, we have extensive evidence for a causal link between OSA and cardiovascular disease (*Gozal and Kheirandish-Gozal, 2008*). Here, we provide definitive evidence of elevated plasma TSP1 level in both humans with OSA and mice exposed to IH. Moreover, we found that TSP1 activated TGFβ signaling, which subsequently promoted the transformation of CFs to myofibroblasts, dependent on STAT3 Tyr705 phosphorylation. Finally, pharmacological inhibition of STAT3 with S3I-201 or AAV9-periostin promoter-shStat3 significantly attenuated IH-induced cardiac fibrosis under both basal and Ang II infusion conditions (*Figure 6H*).

TSP1 belongs to the thrombospondin family, a conserved family of extracellular, oligomeric, multidomain, calcium-binding glycoproteins that can be secreted by various cell types (*Crawford et al., 1998*; *Adams and Lawler, 2011*; *Hugo et al., 1998*). Increased TSP1 levels are associated with many kinds of cardiovascular diseases, including pulmonary hypertension (PH), idiopathic interstitial pneumonia and aging (*Ide et al., 2008*; *van Almen et al., 2011*; *Kaiser et al., 2016*; *Kumar et al., 2017*; *Rogers et al., 2017*). In individuals with end-stage PH, both protein and mRNA levels of TSP1 are elevated in the lung parenchyma and pulmonary artery (*Rogers et al., 2017*). Furthermore, TSP1 blockade protected against Schistosoma- and hypoxia-induced PH (*Kumar et al., 2017*). In our study, TSP1 protein expression was significantly elevated in patients with OSA as compared with healthy controls. Strikingly, OSA incidence is high in patients with PH, idiopathic pulmonary fibrosis, and other cardiac diseases, which suggests that TSP1 might be the common pathologic mechanism in OSA-associated cardiovascular diseases. Therefore, TSP1 might be a biomarker for diagnosis of OSA-related cardiac dysfunction, and reducing IH-upregulated TSP1 expression might be crucial to protect the cardiac function of patients with sleep apnea.

TGFβ, perhaps the most extensively studied mediator of fibroblast activation and having the greatest role in pathological fibrosis (*Travers et al., 2016*; *Leask and Abraham, 2004*), can be activated by TSP1 binding to a defined site on latency-associated propeptide and inducing a conformational change in the latent complex (*Murphy-Ullrich and Poczatek, 2000*; *Ribeiro et al., 1999*). The kinetics of its gene expression, along with its independence from de novo protein synthesis, led to classifying TSP1 as an immediate early-response gene (*Majack et al., 1987*; *Patel et al., 1997*). For example,in vascular smooth muscle cells, the mRNA level of TSP1 was induced rapidly within 15 min by platelet-derived growth factor (*Majack et al., 1987*). Consistently, we found that IH upregulated *Thbs1* continuously from 2 to 8 hr. However, IH-induced TSP1 protein expression in PCFs peaked at

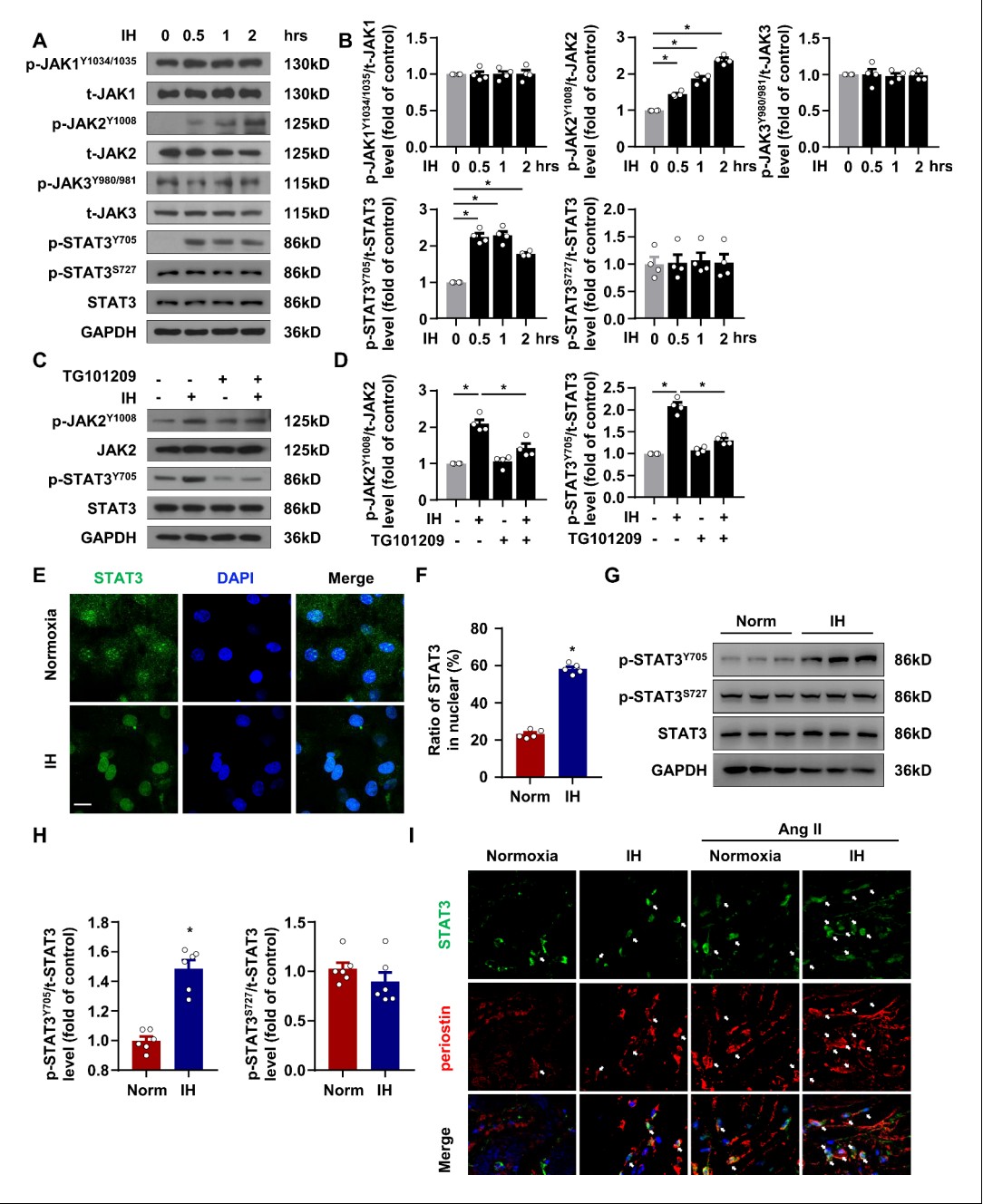

**Figure 4.** IH induced STAT3 signaling activation in CFs. (**A**) mPCFs were cultured and exposed to IH for the indicated time. The protein levels of phosphorylated JAK1 at Tyr1034/1035 (p-JAK1$^{Y1034/1035}$), total JAK1 (t-JAK1), p-JAK2 at Tyr1008 (p-JAK2$^{Y1008}$), total JAK2 (t-JAK2), p-JAK3 at Tyr980/981 (p-JAK3$^{Y980/981}$), total JAK3 (t-JAK3), p-STAT3 at Tyr705 (p-STAT3$^{Y705}$) and Ser727 (p-STAT3$^{S727}$) and total STAT3 (t-STAT3) detected by western blot analysis. (**B**) Quantification of indicated protein levels in (**A**). Data are mean ± SEM, n = 4 independent experiments, *p<0.05, 1-way ANOVA with Bonferroni post-test. (**C–D**) mPCFs were cultured and exposed to IH for 1 hr with or without TG101209 (2 μM). Indicated protein levels were analyzed (**D**). (**E**) mPCFs were exposed to IH for 1 hr. Representative confocal microscopy images of immunofluorescence staining for STAT3 and DAPI. Scale bar, 20 μm. (**F**) Quantification of subcellular localization of STAT3 in (**E**). Data are mean ± SEM, n = 5 independent experiments, *p<0.05, unpaired 2-tail t test. (**G**) The protein levels of p-STAT3$^{Y705}$ and p-STAT3$^{S727}$ in ventricles of mice with or without IH detected by western blot analysis. (**H**) Quantification of p-STAT3$^{Y705}$, p-STAT3$^{S727}$ and t-STAT3 in (**E**). Data are mean ± SEM, n = 6 mice per group, *p<0.05, unpaired 2-tail t test. (**I**) Representative confocal microscopy images of immunofluorescence staining for STAT3, periostin, and DAPI in mice exposed IH

*Figure 4 continued on next page*

*Figure 4 continued*

with or without Ang II infusion for 4 weeks. The white arrows indicate periostin-positive cells with nuclear localization of STAT3. Scale bar, 10 μm. Data used for quantitative analyses as well as the numerical data that are represented in graphs are available in *Figure 4—source data 1*.

The online version of this article includes the following source data for figure 4:

**Source data 1.** Intermittent hypoxia induced STAT3 signaling activation.

2 hr and decreased from 4 to 8 hr. The concentration of TSP1 in culture medium was still increased from 4 to 8 hr, which suggests that IH might promote TSP1 secretion as well. However, the mechanism underlying TSP1 secretion needs further elucidation.

STAT3, as a multifaceted molecule, plays a central role in organic fibrogenesis and cardiac hypertrophy (*Su et al., 2017*). Hypoxic stress can induce phosphorylation of STAT3 and its simultaneous nuclear translocation (*Noman et al., 2009*). Here we found that STAT3, as a TSP1 transcription factor, was phosphorylated at Tyr705 and translocated into the nucleus after IH exposure, then promoted fibroblast activation. In addition, STAT3-null mice showed severe fibrosis with age (*Jacoby et al., 2003*), but in transaortic constriction-induced cardiac remodeling, selective inhibition of STAT3 by S3I-201 significantly improved EF and decreased the left ventricle chamber dilation (*Unudurthi et al., 2018*). Consistently, we found that S3I-201 treatment reduced the pathologic upregulation of TSP1 and ameliorated the cardiac fibrosis of IH-exposed mice with or without Ang II infusion. Importantly, these results were further confirmed by using AAV9-periostin promoter-shStat3. Direct blockade of TSP1/TGFβ signaling is a major therapeutic challenge because of its ubiquitous nature and localized signaling effects (*Kumar et al., 2017*). Our results indicated that targeting the pathologic activation of STAT3 might hold promise as a potentially safe approach in preventing IH-induced cardiac fibrosis.

In summary, our study demonstrated that IH upregulated the expression of TSP1 in both mice and humans, and TGFβ pathway activation induced by JAK2/STAT3/TSP1 signaling played a vital role in IH-induced fibroblast activation and cardiac fibrosis. Pharmacological or genetic inhibition of STAT3 might be a potential therapeutic strategy in managing several diverse yet intertwined human pathologies related to IH.

# Materials and methods

## Key resources table

| Reagent type (species) or resource | Designation | Source or reference | Identifiers | Additional information |
|---|---|---|---|---|
| Antibody | anti-thrombospondin-1 (Rabbit monoclonal) | Cell Signaling Technology | Cat# 37879 | WB (1:1000) |
| Antibody | anti-phospho-SMAD2/3 (Rabbit monoclonal) | Cell Signaling Technology | Cat# 8828 | WB (1:1000) |
| Antibody | anti-SMAD2/3 (Rabbit monoclonal) | Cell Signaling Technology | Cat# 8685 | WB (1:1000) |
| Antibody | anti-phospho-JAK1 (Rabbit monoclonal) | Cell Signaling Technology | Cat# 74129 | WB (1:1000) |
| Antibody | anti-phospho-JAK2 (Rabbit monoclonal) | Cell Signaling Technology | Cat# 8082 | WB (1:1000) |
| Antibody | anti-phospho-JAK3 (Rabbit monoclonal) | Cell Signaling Technology | Cat# 5031 | WB (1:1000) |
| Antibody | anti-JAK1 (Rabbit monoclonal) | Cell Signaling Technology | Cat# 3344 | WB (1:1000) |
| Antibody | anti-JAK2 (Rabbit monoclonal) | Cell Signaling Technology | Cat# 3230 | WB (1:1000) |
| Antibody | anti-JAK3 (Rabbit monoclonal) | Cell Signaling Technology | Cat# 8827 | WB (1:1000) |

*Continued on next page*

*Continued*

| Reagent type (species) or resource | Designation | Source or reference | Identifiers | Additional information |
|---|---|---|---|---|
| Antibody | anti-phospho-STAT3 (Tyr705) (Rabbit monoclonal) | Cell Signaling Technology | Cat# 9145 | WB (1:1000) |
| Antibody | anti-phospho-STAT3 (Ser727) (Rabbit polyclonal) | Cell Signaling Technology | Cat# 9134 | WB (1:1000) |
| Antibody | anti-STAT3 (Rabbit monoclonal) | Cell Signaling Technology | Cat# 12640 | WB (1:1000) ChIP (1:50) |
| Antibody | anti-STAT3 (Mouse monoclonal) | Cell Signaling Technology | Cat# 9139 | IF (1:100) FACS (1 µL per test) |
| Antibody | Normal Rabbit IgG | Cell Signaling Technology | Cat# 2729 | ChIP (1 µg per test) |
| Antibody | anti-thrombospondin antibody (Mouse monoclonal) | Abcam | Cat# ab1823 | IF (1:50) FACS (1 µL per test) |
| Antibody | anti-αSMA (Mouse monoclonal) | Abcam | Cat# ab7817 | WB (1:1000) IF (1:100) |
| Antibody | anti-Collagen I antibody (Mouse monoclonal) | Abcam | Cat# ab6308 | WB (1:1000) IF (1:100) |
| Antibody | anti-periostin (Rabbit polyclonal) | Abcam | Cat# ab14041 | WB (1:1000) IF (1:100) |
| Antibody | anti-vimentin (Rabbit monoclonal) | Abcam | Cat# ab92547 | IF (1:500) |
| Antibody | anti-GAPDH (Mouse monoclonal) | Proteintech | Cat# 60004 | WB (1:1000) |
| Antibody | anti-Ki67 (Rabbit monoclonal) | HuaAn Biotechnology Co | Cat# ET1609-34 | IF (1:100) |
| Antibody | anti-mouse CD45-PE Cy7 (Rat monoclonal) | BioLegend | Cat# 103114 | FACS (1 µL per test) |
| Antibody | anti-mouse CD31-PE Cy7 (Rat monoclonal) | BioLegend | Cat# 102523 | FACS (1 µL per test) |
| Antibody | anti-mouse CD11b-PE Cy7 (Rat monoclonal) | BioLegend | Cat# 101215 | FACS (1 µL per test) |
| Antibody | anti-mouse Ter119-PE Cy7 (Rat monoclonal) | BioLegend | Cat# 116221 | FACS (1 µL per test) |
| Antibody | anti-mouse thy1- Alexa Fluor 488 (Mouse monoclonal) | BioLegend | Cat# 202505 | FACS (1 µL per test) |
| Antibody | anti-mouse thy1- Alexa Fluor 488 (Rat monoclonal) | BioLegend | Cat# 105315 | FACS (1 µL per test) |
| Antibody | PE RAT anti-mouse IgG1 | BD Biosciences | Cat# 550083 | FACS (1 µL per test) |
| Antibody | Alex 594-conjugated goat anti-mouse antibody | Thermo Fisher Scientific | Cat# A-11005 | IF (1:200) |
| Antibody | Alex 488-conjugated goat anti-rabbit antibody | Thermo Fisher Scientific | Cat# A-11008 | IF (1:200) |
| Sequence-based reagent | Stat3 siRNA | Santa Cruz Biotechnology | Cat# sc-29494 | |
| Sequence-based reagent | Control siRNA | Santa Cruz Biotechnology | Cat# sc-37007 | |
| Sequence-based reagent | Mouse Acta | This paper | N/A | Sequences in *Supplementary file 1* |
| Sequence-based reagent | Mouse Col1a1 | This paper | N/A | Sequences in *Supplementary file 1* |

*Continued on next page*

*Continued*

| Reagent type (species) or resource | Designation | Source or reference | Identifiers | Additional information |
|---|---|---|---|---|
| Sequence-based reagent | Mouse Postn | This paper | N/A | Sequences in *Supplementary file 1* |
| Sequence-based reagent | Mouse Thbs1 | This paper | N/A | Sequences in *Supplementary file 1* |
| Sequence-based reagent | Mouse Tnfa | This paper | N/A | Sequences in *Supplementary file 1* |
| Sequence-based reagent | Mouse Stat3 | This paper | N/A | Sequences in *Supplementary file 1* |
| Sequence-based reagent | Mouse 18S | This paper | N/A | Sequences in *Supplementary file 1* |
| Peptide, recombinant protein | Angiotensin II | Abcam | Cat# ab120183 | |
| Commercial assay or kit | TSP1 ELISA KIT(Human) | R and D Systems | Cat# DTSP10 | |
| Commercial assay or kit | TSP1 ELISA KIT(Mouse) | Cloud-Clone Corp. | Cat# SEA611Mu | |
| Commercial assay or kit | TUNEL staining kit | KeyGEN BioTECH | Cat# KGA7073-1 | |
| Commercial assay or kit | SimpleChIP Enzymatic Chromatin IP Kit | Cell Signaling Technology | Cat# 9003 | |
| Commercial assay or kit | Eastep Super Total RNA Extraction Kit | Promega Corporation | Cat# LS1040 | |
| Chemical compound, drug | S3I-201 | Santa Cruz Biotechnology | Cat# sc-204304 | |
| Chemical compound, drug | TG101209 | MedChem Express | Cat# HY-10410 | |
| Software, algorithm | Prism version 7.0 | GraphPad Software Inc | https://www.graphpad.com/scientific-software/prism/ | |
| Software, algorithm | ImageJ version 1.52a | NIH | https://imagej.nih.gov/ij/ | |
| Software, algorithm | FlowJo version 10 | Tree Star Inc | https://www.flowjo.com/solutions/flowjo/downloads | |

## Animals

Male C57BL/6J mice at 8–10 weeks of age were used. All mice were housed in a controlled environment ($20 \pm 2°C$, 12 hr/12 hr light/dark cycle) and maintained on a standard chow diet with free access to water. Intermittent hypoxia (IH) was induced as described previously by using an automated system to control ambient oxygen concentration (*Savransky et al., 2007*; *Chen et al., 2005*). Briefly, $O_2$ concentration was decreased to 4–6% approximately every 60 s. IH mice were exposed to IH for 8 hr/day in the light time for 28 days. In the pathologic cardiac fibrosis model, mice were subcutaneously implanted with an osmotic minipump (2004 model, Alzet, CA) containing Ang II (1 mg/kg/day) in saline (0.9% w/v) or an identical volume of saline. After the surgery, all mice were exposed to IH for 28 days. In rescue experiments, S3I-201 (5 mg/kg) was given by intraperitoneal injection every 2 days during or after the IH exposure.

## Adeno-associated virus (AAV) construction and infection

The AAV used in this study was constructed as previously reported (*Piras et al., 2016*). Briefly, AAV9 carrying a periostin promoter driving the expression of shRNA targeting Stat3 (AAV9-periostin promoter-shStat3) was constructed by Shanghai Genechem Co. (Shanghai). The sequences of the shRNAs were for shStat3, GTCACACAGATGAACTTGGTCTTCAGGT and GCATCAATCCTGTGGTATA. The periostin AAV9-periostin promoter-shStat3 or AAV9-periostin promoter-shScramble

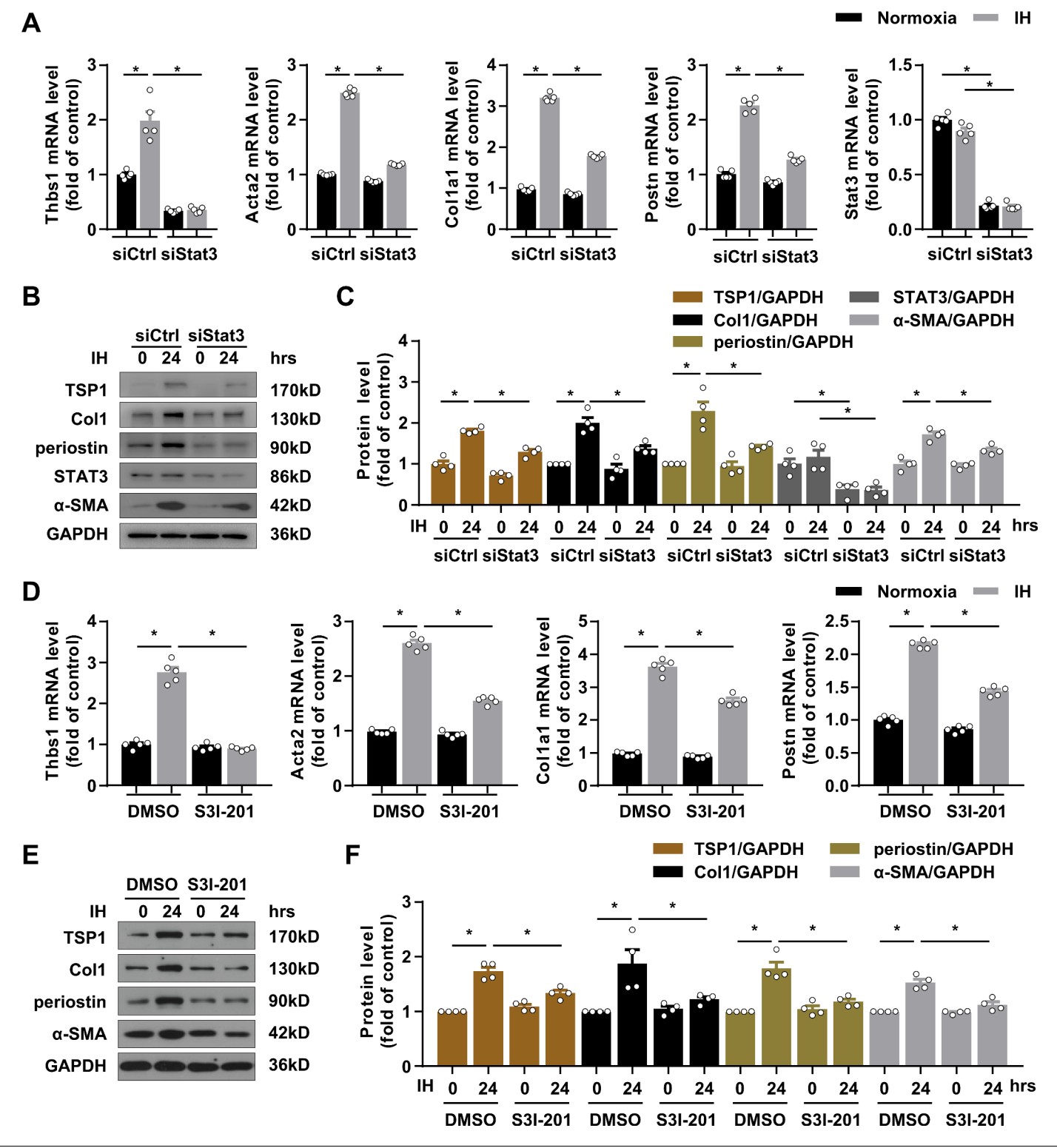

**Figure 5.** Silencing and inactivating STAT3 downregulated TSP1 expression and attenuated IH-induced CF activation. (A–C) mPCFs were transduced with STAT3 siRNA (siSTAT3) or control siRNA (siCtrl) for 24 hr, then cultured under normoxia or IH for 24 hr. (A) *Thbs1*, *Acta2*, *Col1a1*, *Postn*, *Stat3* mRNA levels were quantified by RT-PCR. Data are mean ± SEM, n = 5 independent experiments, *p<0.05, 2-way ANOVA with Bonferroni post-test. (B) The protein levels of TSP1, collagen I (Col1), periostin, STAT3 and α-SMA detected by western blot analysis. (C) Quantification of TSP1, Col1, periostin, STAT3 and α-SMA protein level in (B). Data are mean ± SEM, n = 4 independent experiments, *p<0.05, 2-way ANOVA with Bonferroni post-test. (D–F)

*Figure 5 continued on next page*

*Figure 5 continued*

mPCFs were cultured under normoxia and IH for 24 hr with or without S3I-201 (100 ng/mL). (D) *Thbs1, Acta2, Col1a1,* and *Postn,* mRNA levels detected by RT-PCR and quantified. Data are mean ± SEM, n = 5 independent experiments, *p<0.05, 2-way ANOVA with Bonferroni post-test. (E) The protein levels of TSP1, Col1, periostin, and α-SMA detected by western blot analysis. (F) Quantification of TSP1, Col1, periostin, and α-SMA protein level in (E). Data are mean ± SEM, n = 4 independent experiments, *p<0.05, 2-way ANOVA with Bonferroni post-test. Data used for quantitative analyses as well as the numerical data that are represented in graphs are available in *Figure 5—source data 1*.

The online version of this article includes the following source data and figure supplement(s) for figure 5:

**Source data 1.** Silencing or inactivating STAT3 downregulated TSP1 expression.
**Figure supplement 1.** Chromatin inmmunoprecipitation (ChIP) quantitative PCR assay showed the binding of STAT3 to the Thbs1 promoter.
**Figure supplement 1—source data 1.** Chromatin inmmunoprecipitation (ChIP) quantitative PCR assay.

($1.5 \times 10^{11}$ v.g) was injected into the tail vein of mice. At 1 week post-injection, mice were exposed to IH with or without Ang II infusion.

## Echocardiography

Trans-thoracic echocardiography was performed on all mice by using a Vevo 2100 system with a MS400 linear array transducer (VisualSonics, ON, Canada) as previously reported (*Zhang et al., 2018*). Briefly, mice were anesthetized with 2% isoflurane and kept warm on a heated platform (37° C). The chest hairs were removed by using depilatory cream, and a layer of acoustic coupling gel was applied to the thorax. An average of 10 cardiac cycles of standard 2-D and m-mode short axis at mid-papillary muscle level were analyzed. Left ventricular ejection fraction and dimensions were calculated by using a modified Quinone method.

## Histology

Tissues were fixed in 10% neutral-buffered formalin for 24 hr at room temperature and embedded in paraffin. Hearts were sectioned at 5 μm for staining. Collagen deposition was stained with Masson's trichrome (Sigma-Aldrich, MO) and Sirius red (Solarbio Life Sciences, Beijing) according to the manufacturer's instructions. Images of sections were captured under an Olympus inverted microscope (IX53, Tokyo) and fibrotic areas were semi-quantitatively determined by using ImageJ 1.52.

For immunofluorescence staining, harvested hearts were fixed and embedded in OCT (VWR, PA) and sectioned at 5 μm. Then, sections were stained with primary antibodies for TSP1, Collagen I, periostin, α-SMA and vimentin overnight at 4°C. Alex 488-conjugated goat anti-rabbit and Alex 594-conjugated goat anti-mouse antibodies were used as secondary antibodies. Nuclei were stained with DAPI. TUNEL staining kit (KGA7073-1, KeyGEN BioTECH, Nanjing) was used for analyzing apoptotic cells in cardiac tissue. Images were acquired under an Olympus inverted microscope (IX81, Tokyo). Fluorescent intensity was quantified by using ImageJ. Colocalization of α-SMA and vimentin was quantified by using the Image J with colocalization plugin. Briefly, the plugin initially generates an 8-bit image with only the colocalized points, then it combines the three 8-bit images in an RGB image. Two points are considered as colocalized if their respective intensities are strictly higher than the threshold of their channels.

## RT–qPCR

RNA was extracted by using the Eastep Super Total RNA Extraction Kit (LS1040, Promega, WI). cDNA was synthesized by using the iScript cDNA Synthesis Kit (Bio-Rad, CA). Quantitative RT-PCR was performed with gene-specific primers shown in *Supplementary file 1*.

## Western blot analysis

Proteins were isolated from snap-frozen heart tissue and cultured cardiac fibroblasts (CFs), which were extracted in RIPA solution with a protease inhibitor cocktail (#4693132001) and PhosSTOP (#04906845001, both Roche, IN). Proteins were quantified by using the BCA Protein Assay Kit (Thermo Fisher Scientific, MA). Then, 20 μg each protein was separated on SDS-PAGE and electro-transferred onto PVDF membranes, blocked with TBST containing 5% bovine serum albumin, and blots on membranes incubated with antigen and antibody complexes were detected by an ECL

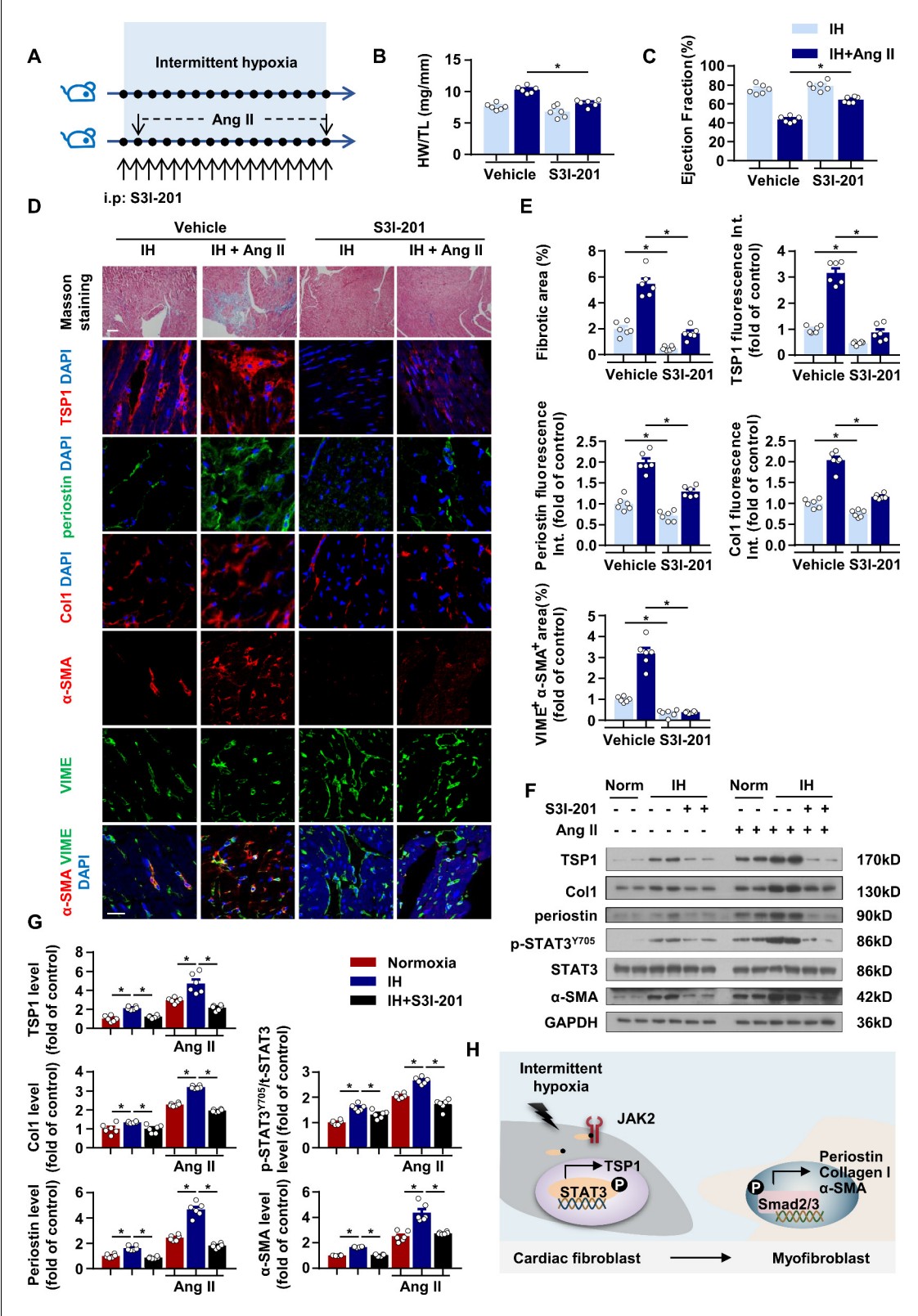

**Figure 6.** STAT3 inhibitor ameliorated IH-induced cardiac fibrosis. (**A**) C57BL/6 mice exposed to IH with or without infusion of Ang II were treated with S3I-201 or vehicle. (**B**) Ratio of heart weight to tibial length of mice. (**C**) Quantification of ejection fraction (EF) after echocardiography of mice. (**D**) Representative Masson staining of left ventricle of mice. Immunofluorescence staining of left ventricle for TSP1, periostin, Collagen I (Col1), α-SMA, VIME and DAPI. Scale bar, 100 μm. (**E**) Quantification of fibrotic area in Masson-stained slides (**D**), quantification of TSP1, periostin, and Col1 fluorescent

*Figure 6 continued on next page*

*Figure 6 continued*

intensity in (D), and quantification of co-localization of α-SMA and VIME in (D). (F) Protein levels of TSP1, Col1, periostin, p-STAT3$^{Y705}$, t-STAT3 and α-SMA in left ventricle of mice detected by western blot analysis. (G) Quantification of TSP1, Col1, periostin, p-STAT3$^{Y705}$ and α-SMA in (F) Data are mean ± SEM, n = 6 mice per group, *p<0.05, 2-way ANOVA with Bonferroni post-test. (H) Schematic diagram depicting the key findings of this study. IH induced cardiac fibrosis via a STAT3/TSP1/Smad pathway. Data used for quantitative analyses as well as the numerical data that are represented in graphs are available in *Figure 6—source data 1*.

The online version of this article includes the following source data and figure supplement(s) for figure 6:

**Source data 1.** Effect of pharmacological inhibition of STAT3 on IH-induced cardiac dysfunction and fibrosis.
**Figure supplement 1.** Genetic inhibition of STAT3 ameliorates IH-induced cardiac dysfunction and fibrosis.
**Figure supplement 1—source data 1.** Effect of genetic inhibition of STAT3 on IH-induced cardiac dysfunction and fibrosis.

protocol with horseradish peroxidase-conjugated IgG as secondary antibodies. Immunoblots were quantified by using ImageJ 1.52.

## Isolation and culture of primary CFs (PCFs)

Isolation of mouse primary CFs (mPCFs) was performed as previously reported (*Su et al., 2017*). Briefly, mouse hearts from freshly euthanized C57BL/6 mice were harvested and minced to 1 mm (*Somers et al., 2008*) in cold phosphate buffered saline. Minced tissue was subsequently digested with buffer containing collagenase II (#V900892, Sigma-Aldrich, MO) and trypsin (Solarbio life Sciences, Beijing) under constant stirring at 37°C for 60 min. The supernatants were spun to collect cells. Then cells resuspended in DMEM/F12 (#11320033, Gibco, MA) were plated into dishes and incubated for 2 hr. Supernatant was discarded and dishes were replenished with fresh medium. mPCFs were incubated at 37°C in a humidified atmosphere of 5% $CO_2$ and grown to 70–80% confluence. Cells at passages 2 to 3 were used in experiments. Cells exposed to IH were maintained in a hypoxia chamber (5% $CO_2$; balance $N_2$ and $O_2$ from 5% to 21%) for the indicated times.

## Flow cytometry

Flow cytometry was performed as previously reported (*Ali et al., 2014*). Digested cells from mouse hearts were suspended in staining buffer containing the relevant surface marker antibodies (PE/Cy7 anti-mouse CD45, CD31, CD11b, Ter119, and Alexa Fluor 488 anti-mouse Thy1) in the dark for 30 min at room temperature. Then, for intracellular staining, cells were fixed with cold methanol for 10 min at −20°C, then incubated with a stain buffer containing TSP1 or STAT3 antibody. After three washing steps, a secondary antibody conjugated to PE was added. The cells were then washed and ready for analysis.

## Cell immunofluorescence staining

Mouse primary CFs were fixed with 4% paraformaldehyde and permeabilized with 0.25% Triton X-100. Then cells were stained with antibodies against α-SMA (1:100), STAT3 (1:100) or Ki67 (1:100). Alexa Fluor antibodies (1:200) were used as secondary antibodies. Stress fibers were stained with rhodamine phalloidin (Thermo Fisher Scientific, MA). Nuclei were stained with DAPI. Images were acquired under an Olympus laser scanning microscope (IX81, Tokyo).

## Collagen gel contraction assay

The collagen contraction assay was performed as previously reported (*Pincha et al., 2018*). Mouse primary CFs were embedded in collagen gel containing collagen type I from rat tail (#08–115, Sigma-Aldrich, MO) and culture medium. Contraction was measured from the gel images after 24 hr.

## Lentivirus construction and infection

Lentiviruses carrying short hairpin (shRNA) for *Thbs1* (LV-shThbs1) and non-specific shRNA (LV-shCtrl) were constructed by Shanghai Genechem Co. (Shanghai). CFs were infected with lentivirus at multiplicity of infection (MOI) 10, and no detectable cellular toxicity was observed.

## ELISA

The secretion of TSP1 in mice or human plasma was quantified by using an ELISA kit (SEA611Mu, Cloud-Clone Corp., TX and DTSP10 R and D Systems, MN) according to the manufacturer's instructions.

## siRNA transfection

Mouse primary CFs were seeded on plates and cultured to 80% confluence, then transfected with Stat3 siRNA (#sc-29494) or control siRNA (#sc-37007, both Santa Cruz Biotechnology, CA) by using Lipofectamine RNAi MAX transfection reagent (Thermo Fisher Scientific, MA) for 24 hr.

## Chromatin immunoprecipitation (ChIP) assay

ChIP was performed as previously reported (He et al., 2018) with SimpleChIP Enzymatic Chromatin IP Kits (Cell Signaling Technology, MA) according to the manufacturer's instructions. Briefly, mPCFs were crosslinked with 1% formaldehyde followed by quenching with glycine for 5 min. Cell lysates were digested by micrococcal nuclease, sonicated, and proteins were immunoprecipitated with antibody to STAT3 or rabbit IgG as a control. After complete washing, immunoprecipitated DNA was eluted with elution buffer and reverse-crosslinked overnight at 65°C. DNA was purified and quantified by quantitative RT-PCR. Enrichment was calculated relative to input. Primers for ChIP-qPCR were forward, 5'- TGGCTTCCTCTGTGGTCTCT-3', and reverse, 5'-GTCAAGGTCATGGGATGGTC-3'.

## Statistical analysis

Sample sizes were designed with adequate power according to the literature and our previous studies. Data are presented as mean ± standard error of the mean (SEM). Statistical analysis involved use of GraphPad Prism 7 v7.04 with two-tailed, unpaired Student's $t$ test or one-way or two-way ANOVA with Bonferroni multiple comparison post-test, as appropriate. The criterion for statistical significance was $p < 0.05$.

# Acknowledgements

This work was supported by grants from the National Natural Science Foundation of China (81800251, 81570304, 81800297).

# Additional information

## Funding

| Funder | Grant reference number | Author |
| --- | --- | --- |
| National Natural Science Foundation of China | 81800251 | Qiankun Bao |
| National Natural Science Foundation of China | 81570304 | Guangping Li |
| National Natural Science Foundation of China | 81800297 | Yue Zhang |

The funders had no role in study design, data collection and interpretation, or the decision to submit the work for publication.

## Author contributions

Qiankun Bao, Resources, Formal analysis, Funding acquisition, Validation, Investigation, Writing - original draft, Writing - review and editing; Bangying Zhang, Kai Zhang, Ming Yuan, Formal analysis, Validation, Investigation; Ya Suo, Formal analysis, Investigation; Chen Liu, Conceptualization, Resources, Investigation; Qian Yang, Resources, Validation, Investigation; Meng Yuan, Resources, Formal analysis, Investigation, Writing - original draft; Yue Zhang, Formal analysis, Funding acquisition,

Writing - original draft; Guangping Li, Conceptualization, Data curation, Supervision, Funding acquisition, Project administration, Writing - review and editing

## Author ORCIDs
Qiankun Bao  https://orcid.org/0000-0002-5221-2780
Chen Liu  http://orcid.org/0000-0001-7120-5626

## Ethics
Human subjects: Ethical approval was obtained through the institutional ethical review board of Peking University People's Hospital (Permit Number: 2018PHB210-01). The study was conducted in accordance with the Declaration of Helsinki. Written informed consent was taken from all study participants.
Animal experimentation: Animal procedures were approved and conducted in accordance with the Experimental Animal Administration Committee of Tianjin Medical University (Permit Number: SYXK 2011-0006; SYXK 2016-0012).

## Decision letter and Author response
Decision letter https://doi.org/10.7554/eLife.49923.sa1
Author response https://doi.org/10.7554/eLife.49923.sa2

## Additional files

### Supplementary files
- Supplementary file 1. RT-PCR primers.
- Transparent reporting form

### Data availability
All data generated or analysed during this study are included in the manuscript and supporting files. Source data files have been provided for Figures 1 to 6.

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
