## [Decision Letter]

**Acceptance summary:**

The paper, which is much improved after extensive revision in response to reviewers' comments, provides a detailed dissection of the impact of intermittent hypoxia (IH), as found in sleep apnea, on fibotic signalling through activation of JAK2, STAT3 and TSP1, leading to expression of SMA, Col1 and periostin reflecting fibroblast activation. A nice feature of this study is that the relatively mild effects of short term IH alone, significantly aggravate the pro-fibrotic effects of Ang II, allowing greater dynamic range for demonstration of its molecular and functional impact on the heart. Significantly, inhibition of STAT3 signaling reverses this impact, pointing to a possible new therapeutic approach for sleep apnea. The study effectively combines in vivo and in vitro evidence for molecular mechanism.

**Decision letter after peer review:**

Thank you for submitting your article "Intermittent hypoxia mediated by TSP1 dependent on STAT3 induces cardiac fibroblast activation and cardiac fibrosis" for consideration by *eLife*. Your article has been reviewed by three peer reviewers, and the evaluation has been overseen by a Reviewing Editor and Jonathan Cooper as the Senior Editor. The following individual involved in review of your submission has agreed to reveal their identity: Barry Fine (Reviewer #2). The reviewers have discussed the reviews with one another and the Reviewing Editor has drafted this decision to help you prepare a revised submission.

Summary:

The study proposes a novel mechanism for the induction of cardiac fibrosis in patients suffering obstructive sleep apnea, which involves body-wide exposure to intermittent hypoxia (IH). A mouse model was established of IH in the context of a pro-fibrotic stimulus provided by Angiotenin II (Ang II). Cardiac functional outputs were explored in the various treatment groups comparing normoxia and IH. Developed fibrosis (sirius red staining) and induction of signaling pathways (western blotting, qRT-PCR, IF) were also explored. Through a series of in vivo and in vitro studies, the authors find that IH enhances the cardiac fibrosis, HW/BW changes and compromised cardiac outputs induced by Ang II. IH alone had weaker and non-significant effects. They define a pathway for the IH/Ang II interaction involving activation of the transcription factor STAT3 (through phosphorylation at position Y705), leading to induction of TSP1 (an inducer of latent TGFβ) and TGFβ/Smad2/3 signaling pathway, the main known inducer of cardiac fibrosis in different injury situations. siRNA and chemical inhibitors of STAT3 and TSP1 were used in vivo and/or in vivo to confirm the pathway links. Inhibition of STAT3 using S3I-201 significantly restored cardiac ejection fraction and reduced scar size, accompanied by changes in the proportion SMA+ fibroblasts, STAT3 activating phosphorylation, TSP1 protein levels.

Essential revisions:

The paper is extensive and novel, and the data are generally clean with a clear message. However the reviewers have identified a number of significant issues that need to be addressed before publication in *eLife*. In particular, the characterisation of fibrosis using just SMA staining or gene expression in vitro and in vivo is insufficient (SMA is a poor marker of myofibroblasts), and the rapid induction of TSP1 and SMA in vitro within 2 hours begs the question of the validity of the mechanism of activation of fibrosis promoted. We would specifically like to see data on the mechanism of activation of STAT3 and via which JAKs if possible, stronger evidence for STAT3 activation in vivo, and determination that TSP1 is a direct target of STAT3. Strengthening the assessment of fibrosis and accompanying inflammation could be achieved by analysis of collagen and other ECM components directly, and inflammatory markers. Use of genetic markers such as Postn-Cre would be desirable but not essential. Also, many of the other questions and comments should be addressed, although we don't consider human data to be an essential revision.

1) The concept of Stat3-TSP1-TGFβ signaling pathway in cardiac fibrosis activation is interesting. However, the traditional signaling involved in ischemia/hypoxia mediated cellular signaling often leads to cellular injury and death. This study offers a potential opportunity to illustrate if Stat3 activation in cardiac fibroblasts is a specific and unique signaling event for IH vs. ischemia/hypoxia injury. It is also important to investigate the molecular mechanism involved in Stat3 activation under IH and Ang II stimulation. Which JAK molecule is involved? How was Stat3 selected to study among other possible signaling pathways?

2) Cellular specificity of the Stat3 signaling in cardiac fibrosis should be better demonstrated. Stat3 mediated signaling has a significant impact on cardiomyocyte hypertrophy and survival. Much of the in vivo characterization, including Stat3 measurement or inhibitor treatment, can potentially be contributed by myocytes rather than fibroblasts. Their interaction could also be an important contributor to the hypertrophic remodeling and contractile dysfunction. Therefore, authors should recognize the limitation of these data and their in vivo experimental approaches. They should carefully determine the myocyte morphology, apoptotic status and function in conjunction with fibrosis analysis. It is also important to recognize that the IH effect is global to all tissues. Overall, the lack of fibroblast specific manipulation of Stat3 or TSP1 in vivo appears to be a significant limitation of the current study.

3) In characterization of the relevance of TSP1 induction among OSA patients, there was no data demonstrating potential spectrum of cardiac function among the OSA patients (their Echo parameters) which could be a major support for the functional relevance of the elevated plasma TSP1 in the cardiovascular complications among OSA patients.

4) The use of human serum samples is quite intriguing and hypothesis generating and unfortunately not very well developed. Clinical relevance and human applicability is always suspect in terms of mouse models so it is quite refreshing that it appears there is some signal of TSP1 in human serum – though this is not well fleshed out by further experiments. The paper, specifically Figures 1 and 2, are very reliant on finding a phenotype that reaches significance thresholds by overlaying with a model of hypertrophy using angiotensin II. The interaction between OSA and hypertension and heart failure is a very hot topic and I think that the authors here really missed a significant opportunity to make a clinical leap. They showed conclusively an additive interaction between IH and angiotensin. What would have really elevated this paper into clinical relevance is treating their mouse model somewhere with an ARB to see if they can mitigate either some of the signaling or mouse phenotype. That could have then been further investigated in their human samples, specifically the TSP1 levels stratified by whether the patient is on an ACEi or ARB. This could have been a real nidus for a clinical trial in terms of treating OSA induced hypertension with RAAS blockade.

5) The authors use a published intermittent hypoxia model but do not show any indication that the hearts are hypoxic. Based on the mechanism proposed, it is important to distinguish between a direct impact of hypoxia on cardiac fibroblasts (CF) as implied from in vitro data, from myriad other mechanisms that may impact cardiac function including kidney dysfunction and inflammation, to name just two. Showing hypoxia with a hypoxia probe such as pimonidazole is important.

6) The authors have superficially evaluated the impact of IH and Ang II by evaluating only α-SMA expression as a readout of myofibroblast activity. The authors do not evaluate collagen or other ECM genes and do not evaluate the contribution of CF proliferation to the phenotype. Do CF proliferate (e.g. BrdU incorporation, Ki67 or Phospho-histone H3)? Are the CF secreting ECM or other inflammatory cytokines? Are CF contractile? What is the impact on inflammation (TSP1 has been shown to induce inflammation in other heart disease models)? There are also a number of genetic markers of fibroblasts that might complement this study or allow the authors to isolate fibroblast populations by flow cytometry (Tcf21-MerCreMer, Postn-MerCreMer, Col1-CreERT2, Pdgfralpha-MerCreMer).

7) The authors also rely heavily on co-localization of SMA and vimentin and TSP1 and vimentin by Immunofluorescence. How is co-localization defined (the methods only state fluorescence intensity was quantified by image J)? How is a secreted factor (TSP1) co-localized with an intracellular CF marker? There seems to be considerable background expression of TSP1, perhaps in cardiomyocytes, and very little seems to be specific for CF.

8) The authors have not shown the mechanism whereby IH induces Stat3 dependent phosphorylation and activity, nor have they demonstrated Stat3 occupancy of a relevant binding site that regulates Tsp1 gene expression with IH. The authors have also not shown Stat3 nuclear accumulation in IH treated mice in vivo, either by immunostaining or nuclear-cytoplasmic fractionation and Western. Many of the observed results thus could be true-true and unrelated.

9) Figure 3D shows a very rapid induction of Tsp1 and SMA protein following IH (2 hrs). This is extremely rapid to see SMA induction even by exogenous TGF-β administration, which typically works on the order of days. Furthermore, the finding that TSP1 and SMA are both induced at 2 hrs of IH seems to argue against a linear IH-Stat3-Tsp1-SMA/myofibroblast model as proposed. It seems equally plausible that (If SMA is indeed induced within 2 hrs) that Stat3 or another mechanism acts independently of TSP1 on myofibroblast activation.

10) The authors treat mice with a Stat3 inhibitor and show it can block fibrosis and functional decline. Can this drug reverse damage already developed by IH?

11) Figure 2B refers to a significant induction of Tsp1 mRNA by IH alone, which is not true (mRNA is only induced by Ang II alone, or IH plus Ang II). This is surprising, considering the rapid and robust induction of TSP1 protein shown after only 2 hrs of IH in Figure 3D. What is the cause of this discrepancy? Perhaps looking at more acute timepoints would be useful in the mouse model.

---

## [Author Response]

Essential revisions:The paper is extensive and novel, and the data are generally clean with a clear message. However the reviewers have identified a number of significant issues that need to be addressed before publication in eLife. In particular, the characterisation of fibrosis using just SMA staining or gene expression in vitro and in vivo is insufficient (SMA is a poor marker of myofibroblasts), and the rapid induction of TSP1 and SMA in vitro within 2 hours begs the question of the validity of the mechanism of activation of fibrosis promoted. We would specifically like to see data on the mechanism of activation of STAT3 and via which JAKs if possible, stronger evidence for STAT3 activation in vivo, and determination that TSP1 is a direct target of STAT3. Strengthening the assessment of fibrosis and accompanying inflammation could be achieved by analysis of collagen and other ECM components directly, and inflammatory markers. Use of genetic markers such as Postn-Cre would be desirable but not essential. Also, many of the other questions and comments should be addressed, although we don't consider human data to be an essential revision.

We deeply appreciate the editor and reviewers for the comments and suggestions to improve the scientific merit of the manuscript. These comments are valuable for us and we have performed several experiments to address all the concerns. We have added more than 30 novel figure panels that represent new experiments or contain additional experiments. In the revised manuscript, we included more reparative myofibroblast markers (collagen I and periostin). We prolonged the treatment time of IH and looked at more acute timepoints for primary cardiac fibroblast (CF) activation in vitro. Upstream signaling of STAT3 was assessed and we found that JAK2 phosphorylation mediated IH-induced STAT3 activation. Evidence of IH-induced STAT3 activation was also observed in vivo. ChIP assay proved that TSP1 was a direct target of STAT3 in CFs. Assessment of fibrosis and accompanying inflammation were also strengthened with new experiments. Moreover, we determined the effect of AAV9-periostin promoter-shStat3 upon IH-induced fibrosis and further confirmed the vital role of fibroblast STAT3 in the process. Below please find our response to each of these comments.

1) The concept of Stat3-TSP1-TGFβ signaling pathway in cardiac fibrosis activation is interesting. However, the traditional signaling involved in ischemia/hypoxia mediated cellular signaling often leads to cellular injury and death. This study offers a potential opportunity to illustrate if Stat3 activation in cardiac fibroblasts is a specific and unique signaling event for IH vs. ischemia/hypoxia injury. It is also important to investigate the molecular mechanism involved in Stat3 activation under IH and Ang II stimulation. Which JAK molecule is involved? How was Stat3 selected to study among other possible signaling pathways?

We thank the reviewer for recognizing the novelty of our study and the valuable comments. To address the reviewer’s concern about the upstream of STAT3, we detected phosphorylation of JAK1-3 under IH. As shown in Figure 4A-B, we found that IH induced phosphorylation of JAK2 at Tyr1008 site time-dependently without altering the phosphorylation of JAK1 or JAK3. In addition, a JAK2 inhibitor, TG101209, significantly abolished IH-induced STAT3 activation (Figure 4C-D). These results indicated that JAK2 is involved in IH-induced STAT3 activation.

TSP1 has been reported to be transcriptionally regulated by STAT3 in astrocytes (Tyzack et al., 2014). Also, STAT3 has been found involved in other fibrosis diseases including skin fibrosis (Chakraborty et al., 2017) and liver fibrosis (Wang et al., 2018). These studies led us to hypothesize that STAT3 might mediate IH-induced TSP1 expression and cardiac fibrosis. We have cited these studies and added a brief introduction: “TSP1 was reported to be transcriptionally regulated by STAT3 in astrocytes, and emerging evidence has indicated that STAT3 is involved in fibrosis diseases. Here, we hypothesized that STAT3 might play a vital role in IH-induced TSP1 expression and cardiac fibrosis.”

2) Cellular specificity of the Stat3 signaling in cardiac fibrosis should be better demonstrated. Stat3 mediated signaling has a significant impact on cardiomyocyte hypertrophy and survival. Much of the in vivo characterization, including Stat3 measurement or inhibitor treatment, can potentially be contributed by myocytes rather than fibroblasts. Their interaction could also be an important contributor to the hypertrophic remodeling and contractile dysfunction. Therefore, authors should recognize the limitation of these data and their in vivo experimental approaches. They should carefully determine the myocyte morphology, apoptotic status and function in conjunction with fibrosis analysis. It is also important to recognize that the IH effect is global to all tissues. Overall, the lack of fibroblast specific manipulation of Stat3 or TSP1 in vivo appears to be a significant limitation of the current study.

We agree. Indeed, both STAT3 inhibitor treatment and IH challenge are global to all tissues, although we have demonstrated their roles in vitro. As the reviewer suggested, we assessed effects of IH on cardiomyocyte morphology and apoptotic status (Figure 1—figure supplement 1B-C) in the revised manuscript. Nuclear translocation of STAT3 induced by IH was also observed in CFs in vivo (Figure 4I). In addition, we performed new in vivo experiments with AAV9-periostin promoter-shStat3 to knockdown STAT3 in fibroblasts. In line with the STAT3 inhibitor experiments, STAT3 knockdown in CFs significantly attenuated IH and Ang II-induced cardiac fibrosis and improved cardiac function (Figure 6—figure supplement 1F-K). The new added experiments together with other results further demonstrated that CF STAT3 might mediate IH-induced cardiac fibrosis.

3) In characterization of the relevance of TSP1 induction among OSA patients, there was no data demonstrating potential spectrum of cardiac function among the OSA patients (their Echo parameters) which could be a major support for the functional relevance of the elevated plasma TSP1 in the cardiovascular complications among OSA patients.

We thank the reviewer for the valuable comments. It is indeed helpful to analyze whether the plasma TSP1 level is associated with cardiac function of OSA patients. Unfortunately, we did not find echocardiography records for the patients and healthy subjects. Since the plasma concentration of NT-proBNP is raised in cardiac impairment (Clin Endocrinol (Oxf). 1997 9;47(3):287-96.), we determined the NT-proBNP concentration in plasma of patients and analyzed the correlation of NT-proBNP and TSP1 concentrations, instead. As shown in Author response image 1, the two indexes did not show a significant correlation, which might be due to limited sample number. Another reason might be that the NT-proBNP level could be affected by multiple factors. Considering that the uncertain conclusion might mislead readers, we only present the result in the response letter.

**Author response image 1. respfig1:** Correlation between NT-proBNP and TSP1 level. (**A**) Plasma concentration of TSP1 and NT-proBNP in patients with obstructive sleep apnea (OSA), detected by ELISA and fluorescent immunochromatography. Correlation between NT-proBNP and TSP1 level. Pearson r = -0.2, P=0.3846.

4) The use of human serum samples is quite intriguing and hypothesis generating and unfortunately not very well developed. Clinical relevance and human applicability is always suspect in terms of mouse models so it is quite refreshing that it appears there is some signal of TSP1 in human serum – though this is not well fleshed out by further experiments. The paper, specifically Figures 1 and 2, are very reliant on finding a phenotype that reaches significance thresholds by overlaying with a model of hypertrophy using angiotensin II. The interaction between OSA and hypertension and heart failure is a very hot topic and I think that the authors here really missed a significant opportunity to make a clinical leap. They showed conclusively an additive interaction between IH and angiotensin. What would have really elevated this paper into clinical relevance is treating their mouse model somewhere with an ARB to see if they can mitigate either some of the signaling or mouse phenotype. That could have then been further investigated in their human samples, specifically the TSP1 levels stratified by whether the patient is on an ACEi or ARB. This could have been a real nidus for a clinical trial in terms of treating OSA induced hypertension with RAAS blockade.

We thank the reviewer for offering a professional clinical perspective. To investigate whether ACEi or ARB might have an effect on IH-induced TSP1 expression, we first performed in vitro experiments in primary CFs. As shown in Author response image 2, valsartan treatment had little effect on IH-induced TSP1 expression and STAT3 activation. This result might be explained in part by our new added results (Figure 4A-D). We found that JAK2 phosphorylation mediated IH-induced STAT3 activation. Hence, our in vitro results indicated that JAK2/STAT3 signaling rather than Ang II/AT1R signaling might play a predominant role in mediating IH-induced cardiac fibrosis. Although ACEi or ARB treatment in the mouse model might have beneficial effects and showed promise for clinical application, the global effects of RAAS blockade and the direct effect on blood pressure could not provide direct evidence to support our hypothesis. For the above reasons, we would like to focus on the potential targets found in the present study. In light of this, we hope that the reviewer will agree that in vivo analysis of the effect of ACEi/ARB is outside the scope of the current manuscript. Here, in the revised version, we have provided two potential therapeutic strategies for the IH-induced cardiac fibrosisin vivo. First, we used the STAT3 inhibitor as a pharmaceutical therapeutic strategy to rescue the IH-induced cardiac fibrosis. Second, STAT3 was knocked down in CFs by using AAV9-periostin promoter-shStat3. In line with the STAT3 inhibitor, STAT3 knockdown also showed impressive restorative effects in IH-induced cardiac fibrosis and cardiac dysfunction (Figure 6—figure supplement 1F-K).

**Author response image 2. respfig2:** Effect of valsartan on IH-induced TSP1 expression. (**A**) mPCFs were cultured under normoxia and IH for 24 hr with or without S3I-201 (S3I, 100 ng/mL) or valsartan (Val, 200 μM). The protein levels of collagen I (Col1), periostin, and α-SMA were detected by western blot analysis. (**B**) Quantification of Col1, periostin and α-SMA protein level in (**A**). Data are mean ± SEM, n=4 independent experiments, *P<0.05, 2-way ANOVA with Bonferroni post-test.

5) The authors use a published intermittent hypoxia model but do not show any indication that the hearts are hypoxic. Based on the mechanism proposed, it is important to distinguish between a direct impact of hypoxia on cardiac fibroblasts (CF) as implied from in vitro data, from myriad other mechanisms that may impact cardiac function including kidney dysfunction and inflammation, to name just two. Showing hypoxia with a hypoxia probe such as pimonidazole is important.

According to the reviewer’s suggestion, we performed the experiments with pimonidazole. Mice were exposed to IH for 7 days. Pimonidazole was injected at 60 mg/kg body weight. Mice were sacrificed 1 hr after pimonidazole injection. Then heart tissue was fixed and embedded into OCT for immunofluorescence staining. The pimonidazole-positive area was significantly increased under stimulation of IH. Data are shown in Figure 1—figure supplement 1A in the revised manuscript.

6) The authors have superficially evaluated the impact of IH and Ang II by evaluating only α-SMA expression as a readout of myofibroblast activity. The authors do not evaluate collagen or other ECM genes and do not evaluate the contribution of CF proliferation to the phenotype. Do CF proliferate (e.g. BrdU incorporation, Ki67 or Phospho-histone H3)? Are the CF secreting ECM or other inflammatory cytokines? Are CF contractile? What is the impact on inflammation (TSP1 has been shown to induce inflammation in other heart disease models)? There are also a number of genetic markers of fibroblasts that might complement this study or allow the authors to isolate fibroblast populations by flow cytometry (Tcf21-MerCreMer, Postn-MerCreMer, Col1-CreERT2, Pdgfralpha-MerCreMer).

We thank the reviewer for the valuable suggestions. Several new experiments were performed to address the concerns. First, we agree that α-SMA expression as a only readout of myofibroblast activity is insufficient. So, we observed the other two markers (collagen I and periostin) in most experiments for the revised manuscript (Figure 1, Figure 3, Figure 5 and Figure 6). Second, for the in vitro analysis, we observed the expression of collagen I and periostin in CFs (Figure 3I-K), CF proliferation (Figure 3C), CF contraction (Figure 3D) and expression of inflammatory cytokine (Figure 3I). Finally, following the suggestion of the reviewer, we sorted CF populations by flow cytometry and further confirmed the TSP1 upregulation in CFs of IH-treated mice (Figure 2G).

7) The authors also rely heavily on co-localization of SMA and vimentin and TSP1 and vimentin by Immunofluorescence. How is co-localization defined (the methods only state fluorescence intensity was quantified by image J)? How is a secreted factor (TSP1) co-localized with an intracellular CF marker? There seems to be considerable background expression of TSP1, perhaps in cardiomyocytes, and very little seems to be specific for CF.

We thank the reviewer for the comments. We quantified the colocalization of dual-color immunofluorescence images by using Image J with the colocalization plugin. Briefly, the plugin initially generates an 8-bit image with only the colocalized points, then it combines the three 8-bit images in an RGB image. Two points are considered colocalized if their respective intensities are strictly higher than the threshold of their channels. We detailed the quantification method in the subsection “Histology”.

We thank the reviewer for pointing out that TSP1 acts as a secretion protein and its localization with vimentin might not be convincing. In the revised manuscript, we replaced the statistical analysis of colocalization with quantification of TSP1 intensity (Figure 2E-F and Figure 6D-E). In addition, to address the reviewer’s concern about the source of TSP1, we also sorted fibroblast populations by flow cytometry and further confirmed the TSP1 upregulation in CFs of IH-treated mice (Figure 2G).

8) The authors have not shown the mechanism whereby IH induces Stat3 dependent phosphorylation and activity, nor have they demonstrated Stat3 occupancy of a relevant binding site that regulates Tsp1 gene expression with IH. The authors have also not shown Stat3 nuclear accumulation in IH treated mice in vivo, either by immunostaining or nuclear-cytoplasmic fractionation and Western. Many of the observed results thus could be true-true and unrelated.

We thank the reviewer for pointing out these important issues. To address the reviewer’s concerns about STAT3, we performed new experiments for the resubmitted manuscript. First, we detected the upstream factors of STAT3 and demonstrated that JAK2 but not JAK1 or JAK3 mediated IH-induced STAT3 phosphorylation. IH strongly induced phosphorylation of JAK2 at Tyr1008 but barely had an effect on phosphorylation of JAK 1 or JAK3 (Figure 4A-B). Additionally, a JAK2 inhibitor, TG101209, blocked the IH-increased levels of p-JAK2 and p-STAT3 in vitro (Figure 4C-D). These results are consistent with a previous report that JAK2-mediated STAT3 phosphorylation played a critical role in fibroblast activation in systemic sclerosis (Chakraborty et al., 2017.). Second, to investigate whether STAT3 could transcriptionally regulate the expression of TSP1 in CFs directly, we used ChIP assay and found that IH enhanced the enrichment of STAT3 at the TSP1 promoter in PCFs as compared with the control (Figure 5—figure supplement 1A). Finally, we also observed STAT3 subcellular localization in vivo. In line with our in vitro results, IH markedly induced STAT3 nuclear accumulation in CFs (Figure 4I). We believe that these new added results strongly strengthen our hypothesis.

9) Figure 3D shows a very rapid induction of Tsp1 and SMA protein following IH (2 hrs). This is extremely rapid to see SMA induction even by exogenous TGF-β administration, which typically works on the order of days. Furthermore, the finding that TSP1 and SMA are both induced at 2 hrs of IH seems to argue against a linear IH-Stat3-Tsp1-SMA/myofibroblast model as proposed. It seems equally plausible that (If SMA is indeed induced within 2 hrs) that Stat3 or another mechanism acts independently of TSP1 on myofibroblast activation.

We greatly appreciate the reviewer for asking a critical question. To investigate whether IH could really induce primary CF activation rapidly, we detected the expression of collagen I and periostin, which are more promising markers for myofibroblasts, in primary CFs after IH treatment for 2 hr. IH failed to alter their expression (Author response image 3). Therefore, although IH induced a transient change of α-SMA level, IH might not induce primary CF activation in 2 hr. As mentioned by the editor and reviewers, α-SMA might be a poor marker to indicate myofibroblasts. In the revised manuscript, we prolonged the stimulation time and included more reparative CF activation markers (collagen I and periostin). Collagen I and periostin expression was significantly increased from 24 to 48 hr (Figure 3I-K), which was consistent with the results observed in vivo.

**Author response image 3. respfig3:** Effect of Intermittent hypoxia (IH) on the expression of collagan I and periostin. (**A**) Expression of TSP1, collagen I (Col1) and periostin in mPCFs exposed to IH for the indicated time detected by western blot analysis. (**B**) Quantification of TSP1, Col1 and periostin in (**A**). Data are mean ± SEM, n=4 independent experiments, *P<0.05, 1-way ANOVA with Bonferroni post-test.

10) The authors treat mice with a Stat3 inhibitor and show it can block fibrosis and functional decline. Can this drug reverse damage already developed by IH?

The reviewer has pointed out an important question. To investigate whether the STAT3 inhibitor could reverse the damage already induced by IH, we treated mice with IH and Ang II for 4 weeks, then treated these mice with S3I-201 (5 mg/kg) for another 2 weeks. We detected the cardiac function index and fibrosis area. As shown in Figure 6—figure supplement 1A-E, the groups did not significantly differ in the indices, so STAT3 inhibition might not have beneficial effects on damage already induced. One of the possible reasons might be that although fibroblasts are the main effector cells in the pathogenesis of cardiac fibrosis, reversing fibrosis is an active process that may require co-operation of interstitial cell subsets, immune cells, vascular cells, and cardiomyocytes (J Am Coll Cardiol. 2019;73 (18): 2283-2285.).

11) Figure 2B refers to a significant induction of Tsp1 mRNA by IH alone, which is not true (mRNA is only induced by Ang II alone, or IH plus Ang II). This is surprising, considering the rapid and robust induction of TSP1 protein shown after only 2 hrs of IH in Figure 3D. What is the cause of this discrepancy? Perhaps looking at more acute timepoints would be useful in the mouse model.

Thanks for the suggestion. IH-induced TSP1 expression is a rapid process and we may have missed the peak time point to observe the change of TSP1 mRNA level. To test this possibility, we treated mice with IH and Ang II for 2 weeks and determined mRNA level of TSP1 in heart. As shown in Figure 2B, *Thbs1* mRNA was significantly induced by IH alone, which was consistent with our in vitro results.